# Validation of the Portuguese Version of the Short-Form Glasgow Composite Measure Pain Scale (CMPS-SF) According to COSMIN and GRADE Guidelines

**DOI:** 10.3390/ani14060831

**Published:** 2024-03-08

**Authors:** Mayara T. de Lima, Pedro H. E. Trindade, Renata H. Pinho, Alice R. Oliveira, Juliana Cannavan Gil, Thayná R. Almeida, Nadia C. Outeda, Stelio P. L. Luna

**Affiliations:** 1Department of Anesthesiology, Medical School, São Paulo State University (UNESP), Botucatu 18618-687, Brazil; mayara.travalini@unesp.br (M.T.d.L.); pedro.trindade@unesp.br (P.H.E.T.); 2Department of Population Health and Pathobiology, College of Veterinary Medicine, North Carolina State University (NCSU), Raleigh, NC 40676659896, USA; 3Faculty of Veterinary Medicine, University of Calgary, 3280 Hospital Dr NW, Calgary, AB T2N 4Z6, Canada; renata.pinho@ucalgary.ca; 4Department of Anatomy, Pathology and Veterinary Clinics, School of Veterinary Medicine and Animal Science, Federal University of Bahia, Salvador 40231-060, Brazil; alice.oliveira@unesp.br; 5Independent Researcher, Av Omar Daibert, 01 casa 171 D, São Bernardo do Campo 09820-680, Brazil; julianagilvet@gmail.com; 6Veterinary Hospital, Anclivepa College, São Paulo 03077-000, Brazil; thaaynaraaineri@gmail.com; 7Department of Pharmacology, Faculty of Veterinary Medicine, University of the Republic Montevideo, Montevideo 13000, Uruguay; nadiacrosi@gmail.com; 8Department of Veterinary Surgery and Reproduction, School of Veterinary Medicine and Animal Science, São Paulo State University (UNESP), Botucatu 18618-681, Brazil

**Keywords:** acute pain, behavior, dogs, pain measurement, reliability, validation study

## Abstract

**Simple Summary:**

Pain in animals is assessed by behavioral alterations presented on scales. We aimed to validate the CMPS-SF according to the COSMIN and GRADE guidelines, used to assess the validity and reliability of health instruments. Four trained evaluators assessed 208 videos in four periods (pre-operative, peak of pain, 1 h after peak of pain and analgesia, and 24 h after surgery) of 52 dogs, divided into negative control (animals without pain), soft tissue surgeries, and orthopedic surgeries. The videos were evaluated in two stages, with intervals of 21 days. The CMPS-SF is a unidimensional scale with good reliability both intra-observer and between different observers, and it showed a strong correlation with other scales. Pain scores were higher after than before surgery, reduced after analgesia, and reduced even further after 24 h, which demonstrates its responsiveness. Postoperative pain scores were higher in dogs undergoing surgery versus controls. The scale is homogeneous, and there is a good interrelationship between its items. The detection values for dogs truly with and without pain were 74–83% and 78–87%, respectively. The score for indicating analgesia was ≥5 and the GRADE classification was high, confirming the validity of the scale.

**Abstract:**

We aimed to validate the CMPS-SF according to COSMIN and GRADE guidelines. Four trained evaluators assessed 208 videos (pre-operative-M1, peak of pain-M2, 1 h after the peak of pain and analgesia (rescue)-M3, and 24 h post-extubation-M4) of 52 dogs, divided into negative controls (n = 10), soft tissue surgeries (n = 22), and orthopedic surgeries (n = 20). The videos were randomized and blinded as to when they were filmed, and were evaluated in two stages, 21 days apart. According to confirmatory analysis, the CMPS-SF is a unidimensional scale. Intra-observer reliability was between 0.80 and 0.99 and inter-observer reliability between 0.73 and 0.86. Criterion validity was confirmed by the correlation between the CMPS-SF and other unidimensional scales (≥0.7). The differences between the scores were M2 ≥ M3 > M4 > M1 (responsiveness), and the scale presented construct validity (higher postoperative pain scores in dogs undergoing surgery versus control). Internal consistency was 0.7 (Cronbach’s α) and 0.77 (McDonald’s ω), and the item-total correlation was between 0.3 and 0.7, except for “A(ii)—Attention to wound”. Specificity and sensitivity were 78–87% and 74–83%, respectively. The cut-off point for rescue analgesia was ≥5 or ≥4 excluding item B(iii) mobility, and the GRADE classification was high, confirming the validity of the scale.

## 1. Introduction

It is essential to diagnose and treat pain in animals to ensure well-being and quality of life. Failure to diagnose pain can lead to oligoanalgesia [1]. On the other hand, states of sedation and dysphoria can lead to a false positive diagnosis of pain, causing animals to receive unnecessary analgesia [2,3,4].

Behavioral scales are simple tools for evaluating pain in animals, replacing the verbal expression used as the gold standard in humans. Pain-behavioral scales are non-invasive or intrusive, do not require physical restraint, are cost-free, provide immediate results, and do not require equipment [5,6,7].

The current pain scales that present some level of validation in dogs are the Glasgow, Melbourne (UMPS) [8], and 4A-Vet [6]. The Glasgow Composite Measure Pain Scale (CMPS) [9] and its Short-Form (CMPS-SF) are those with the highest level of validation in relation to the others. The CMPS-SF [10] presents partial content validity, fair to good inter-observer reliability, criterion validity, responsiveness, sensitivity and specificity, and a cut-off point defined as the median among three hospitals. Although the scale presents these attributes and is the most commonly cited scale in the literature [11], it still requires additional validation to meet the criteria recommended by the COnsensus-based Standards for the selection of health Measurement INstruments (COSMIN) [12,13] and by GRADE (Grading of Recommendations, Assessment, Development, and Evaluations), a universal method that grades the quality of evidence and the strength of its recommendations, implemented by the World Health Organization (WHO) [13,14,15]. Both criteria have already been used to evaluate the quality of psychometric instruments in the area of pain in humans and define the level of scientific evidence [12,13,14], and according to them, the CMPS-SF presents a moderate level of evidence [6].

In order to contribute to a more robust psychometric validation for the CMPS-SF, the objective of the current study was to implement all the steps required by COSMIN [12,13] and GRADE [14] using the Portuguese version of CMPS-SF. For this purpose, the intra- and inter-observer reliability of the total sum and each item of the scale, criterion and construct validities, responsiveness, multiple association, item-total correlation, internal consistency, sensitivity and specificity, and the cut-off point defined by the Youden index and its diagnostic uncertainty zone were established.

## 2. Materials and Methods

The project was approved by the Ethics Committee on the Use of Animals (CEUA 0158/2017) of the Faculty of Veterinary Medicine and Animal Science (FMVZ), São Paulo State University (UNESP), São Paulo state, Botucatu campus, and followed the COSMIN and GRADE (Appendix A) checklist and terminology guidelines [13] to ensure methodological quality [11,12], as well as the Animal Research Reporting In Vivo Experiments (ARRIVE 2.0) guidelines (Appendix A) [16].

### 2.1. Experimental Design

This was a clinical, pragmatic, blind, opportunistic, and prospective cohort study, carried out at the experimental kennel facilities of FMVZ at UNESP, Botucatu, Brazil.

### 2.2. Animals

The study included dogs (*Canis lupus familiaris*) of different breeds of both sexes, divided into three groups: negative control group (unneutered dogs not submitted to surgery and considered healthy according to clinical examination) and two surgical groups of orthopedic and soft tissue surgeries.

As inclusion criterion, dogs classified between ASA I and III requiring general anesthesia for surgery were admitted. All dogs were subjected to physical and laboratory examinations (complete blood and serum biochemistry: alanine aminotransferase, urea, creatinine, alkaline phosphatase, gamma-glutamyl transferase, albumin, and globulin). As previous studies have reported that behavior may represent a confounding factor in pain assessment [17,18,19], dogs that did not exhibit fear (tremors, escape response, animals that remained motionless) or aggression (growling, raised hair, attacks or bites) during the anamnesis and behavioral assessment were admitted.

The dogs were included in the study after signature of the Free and Informed Consent Form by the person responsible.

Exclusion criteria were dogs that required continuous daily use medication for cardiac or hemodynamic diseases, neurological alterations (except spinal disorders), urethral/ureteral obstruction, acute or chronic kidney disease, liver disease, dogs treated with medication for chronic pain, and those that presented complications within 24 h after surgery.

### 2.3. Facilities and Environment

Patients were housed individually in 4 × 3 m masonry kennels, with a floor covered with rubber mats and a 140 × 80 × 80 cm stainless steel cage. The cage was placed in the kennel to follow the same methodology described previously [10,20,21,22]. The dogs were housed in the experimental kennel 24 h before the start of filming to acclimatize them to the location, the researcher, the use of surgical clothing, and/or dressings and adhesive bandages in the region of the cephalic or lateral saphenous vein. These were included from the beginning to blind the identification of perioperative periods during filming. Water and food were available during all filming periods to avoid identification of the pre-operative fasting period. 

### 2.4. Perioperative Period

All dogs, including the control group, were fasted preoperatively for eight hours. Food was withdrawn after the baseline videos were recorded, and water was not available for 2 h before anesthesia. On the day of surgery, sedatives and/or analgesics were administered, according to the medical team’s decision. Next, antisepsis was performed on the surgical regions and the venipuncture area. Dogs undergoing surgery (orthopedic and soft tissue surgery) received anti-inflammatory medication and antibiotics. During the intraoperative period, blood pressure was measured via ultrasonic Doppler (Parks medical 811-b; Parks Medical Electronics Inc., Aloha, OR, USA) or invasively by intra-arterial catheterization, as well as heart and respiratory rate, ECG, capnography, pulse oximetry, and rectal temperature every five minutes (multiparametric monitor-Lifewindow LW9xVet; Digicare Animal Health, Boynton Beach, FL, USA).

### 2.5. In-Person Assessment of Perioperative Pain

The in-person observer (MTL), blinded to protocols and intraoperative information, evaluated the dogs immediately before sedation and every hour until 8 h and 24 h after extubation using the Abbreviated Sedation Scale (0 to 12) [23] and the CMPS-SF [10].

Initially, the dogs were observed inside the kennels, without interaction, for 30 s. Then, the animal’s behavior, interaction in the presence of the evaluator, interest in the environment, posture, and comfort were evaluated as follows: “A(i)—Spontaneous behavior” and “A(ii)—Attention to wound”; the cages were then opened to assess “C(iv)—Response to touch” and “B(iii)—Mobility”. Additionally, behavior and posture were evaluated in general [“D(v)—Demeanor” and “D(vi)—Posture”].

### 2.6. Rescue Analgesia

When a score ≥ 6/24 or ≥5/20 was observed (for orthopedic cases, when excluding the item “B(iii)—mobility”) for the CMPS-SF sum, the first analgesic intervention was performed with 0.5 mg/kg of morphine (Dimorf^®^, Cristália, Itapira, São Paulo, Brazil) associated with methamizol (Analgex V^®^, Agener União, São Paulo, Brazil) (25 mg/kg), intramuscularly (IM), except when the sedation score was ≥6/12. In this case, a new assessment was performed 60 min later.

One hour after rescue analgesia, the dog was re-evaluated, and if the scores reached the cut-off points, the second rescue was administered with 0.5 mg/kg of morphine associated with 1 mg/kg of ketamine (Dopalen^®^, Ceva, Paulínia, São Paulo, Brazil), IM. If a third rescue analgesia was necessary, an additional 1 mg/kg of IM ketamine was administered, and the total number of rescue analgesics was recorded for each dog.

### 2.7. Translation, Back-Translation, and Semantic Equivalence

A native Brazilian fluent in English translated and a native English person fluent in Portuguese back-translated the CMPS-SF and sedation scale [10,23]. A new translation was carried out following the European certification UNE-EN-15038:2006 [24] (European standard for provision of translation services) (Appendix A).

### 2.8. Filming, Editing, Selection, and Evaluation of Videos

The dogs were filmed for approximately three minutes at each assessment period by a camera (GoPro Hero Black 5+; San Mateo, CA, USA) positioned on a tripod one meter away from the cage, in the presence of the responsible researcher (MTL), at baseline (M1-baseline) (before food deprivation) and 1, 2, 3, 4, 8, and 24 h after extubation.

The time point of greatest pain intensity (M2-pain) was defined according to the highest CMPS-SF score obtained by in-person assessment between 1 and 8 h post-extubation; M3 (rescue) corresponded to 1 h after the first rescue analgesia and peak of pain, and M4 (24 h) corresponded to 24 h after extubation. A total of 208 videos (4 time points × 52 animals) were analyzed.

The videos were edited to 90 s video clips that proportionally represented the behaviors observed in the original three minutes. The videos were randomized by the Microsoft Office Excel function = RANDOMBETWEEN (1;208) and evaluated blindly by four observers. 

After watching the videos, the observers responded whether they would perform rescue analgesia and completed the visual analogue scale [VAS—0 (no pain) to 100 mm (worst possible pain)], numerical rating scale [NRS—0 (no pain) to 10 (worst possible pain)], simple descriptive scale [SDS—0 (no pain) to 4 (worst possible pain)], Abbreviated Sedation Scale [0 (awake) to 12 (highest level of sedation)] [23], and the CMPS-SF [10].

### 2.9. Evaluators and Training

Four female evaluators, including an MSc student responsible for in-person evaluations (MTL), a PhD student (ARO), both with residence in Veterinary Anesthesiology, a veterinary ethologist (JCG), and a 3rd year veterinary medicine student (TRA) evaluated the videos.

The training consisted of reading the instructions and a video tutorial reporting how to apply the scales. Next, videos were shared with demonstrative behaviors for each item and sub-item of the CMPS-SF [10] (www.animalpain.org, accessed on 11 September 2020).

### 2.10. Official Assessment

The four evaluators assessed the videos in a randomized and blinded manner regarding time points and groups, twice, with an interval of 21 days between assessments to prevent observers from remembering the first answer.

### 2.11. Statistical Description

For reliability studies, the sample size should be at least 30, using a minimum of three evaluators [25]. According to COSMIN, a minimum sample number of seven times the number of items on the scale is required [26]. Considering that the CMPS-SF contains six items, the sample size required was 42 dogs. Additionally, to confirm the power of the test, it was considered that 75% of the animals undergoing surgery would present a CMPS-SF score ≥ 6 after surgery, while those in the control group would present a score < 6. According to the chi-square test, eight animals were needed in each group for an alpha of 0.05 and test power of 80%.

Statistical analyses were performed with the R software in the Rstudio integrated development environment (Version 1.0.143-© 2009–2016, R. RStudio, PBC, Boston, MA, USA, http://www.rstudio.com/). The IBM SPSS Statistics (Version 20.0, 2011, IBM Corp, Armonk, NY, USA) was used for the chi-square test and Receiver Operating Characteristic (ROC) curve.

The analysis of histograms and boxplots, together with the Shapiro–Wilk test confirmed that the data did not adhere to the Gaussian distribution; thus, non-parametric tests were adopted. Differences were considered significant when *p* < 0.05%. For scale validation analyses, the scores recorded by all evaluators at all times in both phases were used.

#### 2.11.1. Number of Rescue Analgesics

To confirm the difference in pain intensity between the orthopedic and soft tissue groups, the chi-square test was implemented during the in-person assessment carried out by the researcher responsible for this work.

#### 2.11.2. Intra-observer (Repeatability) and Inter-Observer (Reproducibility) Reliability

For repeatability, the results of the first and second phases of evaluation by the same observer were compared. For reproducibility, scores were compared between the four evaluators using both phases. The intraclass correlation coefficients (ICC), two-way random effects model, type agreement, among multiple observers/measurements, and its 95% confidence interval (CI) for the VAS and CMPS-SF total score, or weighted Kappa (Kw) for dichotomous and ordinal variables and their CI (need for rescue analgesia, NRS, SDS, and separate CMPS-SF items) were calculated.

#### 2.11.3. Distribution of Scores

Frequency distribution graphs in percentages were constructed for each item of the CMPS-SF scale at each time point and at all grouped time points to evaluate the representativeness of the items (descriptive analysis).

#### 2.11.4. Multiple Association and Dimensional Structure

The multiple association between CMPS-SF items was analyzed and the scale’s dimension structure was defined in three steps. First, Horn’s parallel analysis was used to decide how many principal components (dimensions) should be retained [27]. Next, a principal components analysis was conducted to analyze the multiple association between the CMPS-SF items in the main components indicated in the previous step. Finally, the fit of the dimensional structure indicated in the first step was examined by confirmatory factor analysis, through the Comparative Fit Index, Tucker-Lewis index (0–1, expecting values > 0.95), and Root Mean Square of Approximation (expecting values < 0.05).

#### 2.11.5. Criterion Validity

As there is no scale considered the gold standard for evaluating acute pain in dogs, the results of the CMPS-SF were compared with the unidimensional scales VAS, simple descriptive scale, and numerical rating scale, as carried out for other species [28,29]. Spearman correlation coefficient interpretation was <0.19 very weak, 0.2–0.39 weak, 0.4–0.59 moderate, 0.6–0.79 strong, and 0.8–1 very strong [30].

#### 2.11.6. Responsiveness

The scores for each item, the sum of the CMPS-SF, and the need for rescue analgesia over time were compared for each group (CG, OG, and STG). 

To sum the CMPS-SF, a negative binomial mixed model was used. For the CMPS-SF items, generalized linear mixed models were applied, adjusted according to Poisson distribution. Evaluators, groups (CG, OG, and STG), phases, time points, and their evaluation order were included as fixed effects and dogs, evaluators, and groups were considered as random effects. For all models, the Bonferroni test was used to adjust the multiple comparisons in the post hoc test.

#### 2.11.7. Construct Validity

Construct validity for the CMPS-SF was determined by testing three hypotheses: (1) if the scale truly measures pain, the peak of pain score (M2-pain) will be higher than the preoperative score (M1-baseline), (2) the score should decrease after rescue analgesia (M2-pain > M3-rescue), and (3) pain scores should reduce over time (M1-baseline < M4-24 h < M3-rescue < M2-pain). A second test was the Known-group validity, in which scores should be the same in the three groups at baseline and higher in orthopedic and soft tissue groups in relation to control group at M2-pain and M3-rescue. 

To confirm construct validity, correlation between items was applied according to principal component analysis, internal consistency, and item-total correlation.

#### 2.11.8. Internal Consistency

##### Item-Total Correlation

To analyze homogeneity, inflationary items, and the relevance of each item on the scale, each item was compared with the sum of the scale after excluding the evaluated item, using Spearman’s rank correlation. Values between 0.3 and 0.7 were accepted [31].

##### Cronbach’s Alpha and McDonald’s Omega Coefficients

Cronbach’s alpha coefficient (α) and McDonald’s omega (ω) were used to estimate the consistency (interrelationship) of the scores for each item [32]. For coefficient α, the interpretation was 0.60–0.64 minimally acceptable, 0.65–0.69 acceptable, 0.70–0.74 good, 0.75–0.80 very good, and >0.80 excellent [32]. For the coefficient ω, acceptable values were considered between 0.65 and 0.8, and strong evidence, > 0.8 [33].

#### 2.11.9. Specificity and Sensitivity

The perioperative scores of the scales were transformed into dichotomous values (‘0’ absence of the item and ‘1’ presence), and the respective equations were applied. 

Specificity = TN/Total of dogs; in which TN = true negatives (scores representing ‘0’ when dogs should be pain-free, i.e., control group scores). Sensitivity = TP/Total of dogs; where TP = true positives (scores representing pain expression ≥ ‘1’ at the time when the most intense pain would be expected in dogs submitted to orthopedic and soft tissue surgeries-M2-pain). 

For the CMPS-SF total score, the percentage of dogs in the orthopedic group with scores < 5 and ≥5 and in the soft tissue group with scores < 6 and ≥6 before (M1-baseline) and after surgery (M2-pain) were considered to calculate the sensitivity and specificity, respectively. Interpretation: Excellent 95–100%, good 85–94.9%, moderate 70–84.9%, non-specific or non-sensitive < 70% [31].

#### 2.11.10. Determination of Intervention Score for Rescue Analgesia

The indicative score for rescue analgesia was calculated by whether or not analgesia was indicated after the evaluators watched each video, according to their clinical experience. 

The calculation of the area under the curve indicates the discriminatory capacity of the test. The ROC curve and area under the curve graphically represent true positives (sensitivity) and false positives (1-specificity). The Youden index determined by the ROC curve consists of the simultaneous point of greatest sensitivity and specificity. The highest value of the Youden index consists of Youden index = (sensitivity + specificity) − 1 and represents the indication point for rescue analgesia. An AUC ≥ 0.95 indicates the high discriminatory capacity of the scale [34].

The zone of diagnostic uncertainty was determined by calculating the 95% confidence interval of the Youden index, replicating the original ROC curve 1001 times using the bootstrap method.

## 3. Results

The study initially included 54 dogs of different breeds (Figure 1). Two dogs were excluded, one due to aggressiveness after extubation and the other due to postoperative hemorrhage and the requirement for additional surgery. The 52 remaining dogs composed the three groups: ten unneutered female (n = 8) or male (n = 2) dogs made up the negative control group (CG), with a mean age of 3 ± 2 years (1–5) and weight of 14 ± 13 kg (1–27); and forty-four female (n = 27) or male (n = 17) dogs, with a mean age of 5 ± 5 years (0.5–10 years) and weight of 14 ± 10 kg (4–24 kg), admitted to the routine care at the Veterinary Teaching Hospital of FMVZ to undergo orthopedic (OG; n = 21) or soft tissue surgeries (STG; n = 23) made up the other two groups (Appendix A).

### 3.1. Number of Rescues Administered

There was no difference between the number of rescues administered between the orthopedic group (18 dogs of 20) and dogs undergoing soft tissue surgery (18 of 22) (*p* = 0.9).

### 3.2. Reliability

#### 3.2.1. Repeatability (Table 1)

The indication for rescue analgesia, simple descriptive scale, numerical rating scale, and VAS showed good to very good repeatability (0.61 and 0.99), based on the confidence interval. The CMPS-SF showed reasonable to very good repeatability (0.24 to 1) for the different items and, for the sum of the items, the intra-observer reliability was good to very good (0.80 and 0.99) (Table 1).

#### 3.2.2. Reproducibility (Appendix A)

The rescue analgesia, simple descriptive scale and numerical rating scale showed fair to good inter-observer reliability (0.22 and 0.67), the VAS showed fair to very good reproducibility (0.3 and 0.81). For each item of the CMPS-SF, the reliabilities ranged from poor to very good (0.11 and 0.88), and for their sum, the reproducibility was good to very good (0.73–0.86). 

### 3.3. Distribution of Scores

Score 0 always predominated before surgery. The presence of scores above 1 increased after surgery, compatible with the presence of pain at M2-pain and M3-rescue, with a tendency for higher scores at M2-pain in relation to M3-rescue. At M4-24 h, there was a similarity of occurrences in relation to M1-baseline, except for item D(v)—Demeanor, in which score 1—“Quiet” predominated at all time points (Figure 2).

### 3.4. Multiple Association and Dimensional Structure

Horn’s parallel analysis indicated retaining only the first principal component from the principal component analysis. All items except A(ii)—Attention to wound presented positive associations with principal component 1 and associations between themselves. According to the confirmatory factor analysis, the Comparative Fit Index and Tucker–Lewis Index values of 0.96 and 0.93, respectively, demonstrate adequacy of the proposed unidimensional structure; however, the Root Mean Square Error of Approximation values were outside what was expected (Table 2 and Table 3, Figure 3). 

### 3.5. Criterion Validity

The concurrent criterion validity was confirmed by the strong correlation between the CMPS-SF sum and the unidimensional scales (≥0.7). The correlation between the unidimensional scales was very strong (>0.8) (Figure 4).

### 3.6. Responsiveness

Over time, the sum of the CMPS-SF for the surgical groups showed the following order of values: M2-pain ≥ M3-rescue > M4-24 h > M1-baseline. In general, the same occurred for the other scales. In the control group, the scores were higher at baseline compared to 24 h only for the CMPF-SF and did not vary between time points for the other scales.

Regarding the difference between the groups, for all scales, the control group scores were lower than those of the surgical groups at all time points, except at baseline, where there was no difference between the control and the soft tissue surgery groups (Figure 5 and Figure 6, Table 4).

Regarding the differences between the groups for each item of the CMPS-SF, in general, the scores were lower in the control group in relation to the surgical groups at the post-operative time points (Table 4).

### 3.7. Internal Consistency and Item-Total Correlation

Internal consistency using Cronbach α was good for the sum of the scale (0.70) and increased when excluding A(ii)—Attention to wound” and “C(iv)—Response to touch” (Table 5). The CMPS-SF presented internal consistency with an acceptable McDonald’s omega. The only item with an increase in McDonald’s omega after its exclusion was A(ii)—Attention to wound.

The item-total correlation for CMPS-SF varied between 0.3 and 0.7, except for “A(ii)—Attention to wound” [31] (Table 5).

### 3.8. Specificity and Sensitivity

Both the sensitivity and specificity of the CMPS-SF were moderate. For its separate items, only Posture did not present specificity, and only Mobility and Demeanor presented sensitivity (Table 6).

### 3.9. Determination of Intervention Score for Rescue Analgesia

The area under the curve showed moderate discriminatory capacity (AUC = 89.43) [34] for CMPS-SF and values > 0.9 for unidimensional scales [34]. For the two surgical groups combined, the cut-off point according to the Youden Index is ≥5 and the zone of diagnostic uncertainty is between 3.5 and 4.5; therefore, scores ≤ 3 indicate truly pain-free dogs and ≥5 indicate dogs truly in pain (Figure 7, Table 7). The pain score calculated for the surgical groups separately was also ≥5 in both cases. Exclusion of A(ii)—Attention to wound and B(iii)—Mobility separately or together or C(iv)—Response to touch reduced the cut-off point of surgical groups’ data together to ≥4.

In the soft tissue group, when item B(iii) mobility or A(ii) attention to wound was excluded, the cut-off point was ≥4, and when both were excluded simultaneously, the cut-off point remained ≥4. In the soft tissue group, exclusion of C(iv)—Response to touch has not modified the cut-off point which was maintained at ≥5. For the orthopedic group, when excluding items, A(ii) and B(iii), the cut-off point was ≥4, and reduced to ≥3 when only C(iv)—Response to touch was excluded; however, in this case, there was a considerable reduction in the AUC interfering in the diagnostic capacity of the scale, showing that this item is important in assessing pain after orthopedic surgery.

The percentage of dogs that presented scores within the zone of diagnostic uncertainty (4) was below 7% at all time points (Table 8).

## 4. Discussion

The current study complemented the validation of the CMPS-SF [9,10,11] to increase the robustness of the most recognized scale in the world [6,11] that assesses acute pain in dogs. We used a blind and randomized evaluation of videos [38], included a negative control group and analyzed the importance of each item on the scale according to the criteria of intra- and inter-observer reliability, responsiveness, sensitivity, specificity, and internal consistency. The multiple association and item-total correction was evaluated for the first time, and the cut-off point for indicating analgesia was defined using the ROC curve and its Youden Index. All these procedures were performed to fulfill the GRADE and COSMIN guidelines. The current version of the scale increased the GRADE level of evidence from moderate [6] to good [12,13,14,15,39]. In addition, the present study validated the CMPS-SF in the Portuguese language, to guarantee the application of the instrument in another cultural context [40].

The CMPS-SF is reliable, accurate, responsive, sensitive, and specific and showed criterion and construct validity. The CMPS-SF showed better reliability for diagnosing pain than unidimensional numerical scales, as demonstrated for the original scale [9]. The cut-off point, which was different in the original article for orthopedic and soft tissue surgeries, was adjusted to a single benchmark of ≥5/20. Although already on the short-form scale [10], it is not recommended that dogs walk in the post-operative period of orthopedic surgeries, or even in the post-operative period of non-orthopedic surgeries; thus, item B(iii)—Mobility could be excluded in cases of impossibility of locomotion [10]. Its assessment is intrusive, interfering with the dog’s well-being, and can lead to false positives in sedated, fearful, shy, or aggressive dogs [41]. 

The repeatability and reproducibility of the CMPS-SF sum were good to very good, which supplies information of intra-observer reliability for the CMPS-SF, which has only been measured in the long form (CMPS) [42]. The intra- and inter-observer reliabilities of the other scales assessed in the current study were slightly lower than the CMPS-SF, as demonstrated in cats, dogs, and rabbits [28,29,42,43], which highlights the advantage of using composite scales instead of unidimensional ones, even by experienced evaluators. The lower reliability of the unidimensional scales may be justified by the fact that the behaviors to be assessed are not previously defined like in the composite scale, as reported in children, for whom doctors tend to underestimate pain compared to parents, through the application of the numerical rating scale [44].

The reproducibility of the CMPS-SF was much higher than that observed among students and veterinary anesthesiologists (Kw: −0.4–0.73) [45] and that obtained in dogs submitted to bone marrow puncture (Kw: 0.37–0.65) [46], a finding that can be partially justified by the statistical method used (ICC vs. Kw). The ICC is the most suitable test for measuring the reliability of the total score, while the weighted Kappa (Kw) is recommended for categorical variables [25]. According to GRADE, the inter-observer reliability for Kw must be at least 0.40 and the ICC ≥ 0.6, which fits our results for the sum of the CMPS-SF. Another possible justification for our good results could be the previous training via www.animalpain.org (accessed on 11 September 2020), as supported by positive effects of training when assessing pain in rats [47,48] and for the Italian validation of the CMPS-SF [49]. In this way, even with distinct levels of expertise, the four evaluators were able to apply the CMPS-SF reliably.

Items A(ii)—Attention to wound, D(v)—Demeanor, and D(vi)—Posture showed poor inter-observer reliability (<0.2) in some cases. This finding may be related to the lack of detailed descriptions, which respond to the original article’s question about the possible need to reincorporate the developed descriptors [9]. They were removed from the CMPS-SF [10] to reduce tool application time. Additionally, the importance of rigorous statistical tests is reinforced to improve the discriminatory capacity of the instrument, such as the following: content validation based on an ethogram as described for scales validated in other species [50,51], incorporation of the content validity index, and the content quality ratio [52].

Regarding the frequency of occurrence of CMPS-SF items, specific pain behaviors prevailed in the postoperative period, except for “A(i)—Spontaneous behavior” and “A(ii)—Attention to wound”, in that at all time points, the score 0 prevailed. For the item “C(iv)—Response to touch”, the score “Look round (1)” presents a similar percentage at all time points including the baseline period, which is consistent with an apparently normal reaction unrelated to pain. Thus, one could replace this description with “turning the head and looking towards the wound”, as proposed by [42] to facilitate the interpretation of the item and increase its specificity. For the item “D(v)—Demeanor”, the behavior “Quiet (1)” is apparently also not very specific for pain, as it is already present in the spontaneous behavior item with a score of 0, may represent a maintenance behavior [20] or even be related to residual sedation after anesthesia-analgesia or time of day, as reported in rabbits [28]. “Nervous or anxious or fearful” behavior could also be associated with other factors, such as the new hospital environment, separation anxiety, and/or changes in diet [19,53] and presented the same frequency of occurrence at all time points.

Principal component analysis measures the dimensionality and multiple interactions between the items of a scale, segregating them according to their dimensions [54]. Although the CMPS-SF is initially two-dimensional, because it was not possible to estimate the parameters for the Akaike and Bayesian information criteria, confirmatory analysis defined it as unidimensional, like with other scales in other species [28]. However, a tool that assesses different components of pain, such as the emotional [A(i) (vocalization)], motivational [D(v) and D(vi)], motor [B(iii)], and sensorial- discriminative [A(ii) and C(iv)] dimensions is biologically multidimensional [28,55]. 

Concurrent criterion validity is determined by the correlation between the proposed scale and a gold standard instrument [31]. In humans, VAS by self-report is considered the gold standard. In animals, as there is no verbalization and given the absence of a gold standard instrument, criterion validity has been alternatively established by correlation with numerical scales [28,29,43]. The correlations between the three unidimensional scales with the CMPS-SF were strong, thus confirming the criterion validity. The current results were superior to previous ones, in which the correlation between the CMPS and the unidimensional scales was moderate [42].

Regarding responsiveness, pain scores did not reduce after analgesia in dogs undergoing orthopedic surgery, and were slightly lower after orthopedic than after soft tissue surgery at the time point of the most expected pain, possibly due to the residual effect of epidural anesthesia and analgesia performed with bupivacaine and morphine in 60% of the dogs and brachial plexus block in one dog in the orthopedic group. However, responsiveness was observed at other time points, i.e., the scale was sensitive to detect moderate pain at 24 h in relation to baseline. The higher scores observed in the control group in baseline versus 24 h is possibly due to the shorter adaptation of the animals to the environment at baseline. Lack of adaptation may further explain why outliers predominated at baseline in all groups and at all time points in the control group, which draws attention to false-positive results in the presence of fear and anxiety [“D(v) Nervous or anxious or fearful = score 3”]. 

The construct validity was confirmed by the higher pre-operative pain score in the orthopedic group in relation to the other groups, considering that these animals already had pain at this time, resulting in especially higher scores in items C(iv) (response to touch) and D(v) (demeanor). As mentioned previously, after surgery, the residual effect of epidural anesthesia and analgesia abated possible differences between the orthopedic and the soft tissue groups. The lowest pain scores in the control group after surgery confirms the construct validity.

The internal consistency of the CMPS-SF was good for Cronbach’s α and acceptable for McDonald’s ω. The internal consistency of item A(ii), attention to the painful area, increased after its exclusion, and its item-total correlation was unsatisfactory [31], which demonstrates that this item contributes little and has little homogeneity with the scale, respectively. In fact, item A(ii) fits into another dimension according to the multiple association test, and its low sensitivity suggests its exclusion if the scale is refined. The remaining items contributed significantly to the homogeneity of the scale (Rho > 0.3).

To be reliable and applicable, a measurement tool must present high specificity and sensitivity in differentiating animals without or with pain, respectively, to provide analgesia only when necessary [28]. The sum of the CMPS-SF showed moderate specificity and sensitivity. However, individually, the items A(i) Spontaneous behavior, A(ii) Attention to the wound, C(iv) Response to touch, and D(vi) Posture did not show sensitivity or specificity. It is possible the lack of robust content validation and the suppression of the description of each item may have compromised the accuracy of these assessments, as highlighted by six diplomates who suggested, “Revisions should be considered of the CMPS to clarify descriptors and remove or modify items that may not be associated with pain in dogs” [42].

The cut-off point was above the zone of diagnostic uncertainty, which guarantees high accuracy in diagnosing the absence or presence of pain. The narrow zone of diagnostic uncertainty guaranteed a very low percentage of scores diagnosed as false positives or negatives, ensuring an optimal discriminatory ability of the scale to avoid unnecessary treatment or oligoanalgesia, respectively. The cut-off point defined by the Youden Index (≥5) was the same regardless of the type of surgery. This result was equal to that of the original article for orthopedic surgeries, but lower than that defined for soft tissue surgeries [10]. The exclusion of B(iii) Mobility has been previously suggested [41], to evaluate sedated animals and avoid the need to handle the animal, which would make the scale less intrusive. The area under the curve indicates only moderate precision for the discriminatory capacity of the CMPS-SF [34]. It is possible to exclude some items of the scale, like A(ii)—Attention to wound and/or B(iii)—Mobility or C(iv)—Response to touch, in case of difficulty in evaluating them, or even considering the refinement of the scale (e.g., A(ii)—Attention to wound did not perform well in the principal component analysis, item-total correlation, internal consistency, and sensitivity). On the other hand, it should be noted that the removal of one or more of these items can reduce the area under the curve and affect the discriminatory capacity of the scale. 

According to our selection criteria, we included only docile dogs since fear, anxiety, and aggression interfered with acute postoperative pain scores in felines [17], and aggressiveness would make it difficult to evaluate the response to palpation and mobility. Although there could be interactions between stress and pain behaviors, the increase in postoperative anxiety levels in dogs did not affect the CMPS-SF scores [19].

The limitations of this study are as follows: videos originating from in-person recordings lasting up to 10 min were reduced to 90 s to minimize evaluators’ fatigue, based on previous methodologies [28,56]. For editing, efforts were made to ensure that the video proportionally represented the behavioral repertoire of that period including proportional duration and frequency. The absence of male evaluators is another limitation, as men tend to underestimate pain scores compared to women [57].

The inclusion of the in-person researcher as an observer may be a bias. Although she might have been able to recognize the group each dog pertained to (control, orthopedic or soft tissue), she was not able to identify which time point she was assessing. This methodology was previously performed in cats [56] and rabbits [28].

The administration of analgesic rescue was intramuscular to minimize morphine-induced histamine release compared to the intravenous route [58].

A confounding factor when using any pain scale is the presence of residual sedation and/or the effect of anesthetic blocks that impede proprioception. The calculation of the cut-off point excluding the assessment of mobility and attention to the affected area makes the scale more versatile and adaptable to the patient’s clinical conditions, as described for the long and short versions of the UFEPS in cats [43,56]. In this way, we speculate it would not be necessary to wait two hours to assess the pain, as suggested in the original article [10].

## 5. Conclusions

The CMPS-SF is practical, reliable, and effective for diagnosing acute pain in dogs. The item-total correlation, internal consistency, responsiveness, specificity, and sensitivity ensured the content, criterion, and construct validity of the Portuguese version of the scale in accordance with COSMIN guidelines. The GRADE method provided evidence that our methodology increased the level of evidence of the scale from a moderate to a good level. The exclusion of items A(ii) or B(iii), reducing the cut-off point to ≥4, did not interfere with its diagnostic accuracy, facilitating the usability of the tool and making it non-intrusive. However, it is necessary to carry out a new statistical analysis excluding such items in order to investigate whether or not there would be an improvement in the performance of the scale as a whole, in the validation criteria used herein.

## Figures and Tables

**Figure 1 animals-14-00831-f001:**
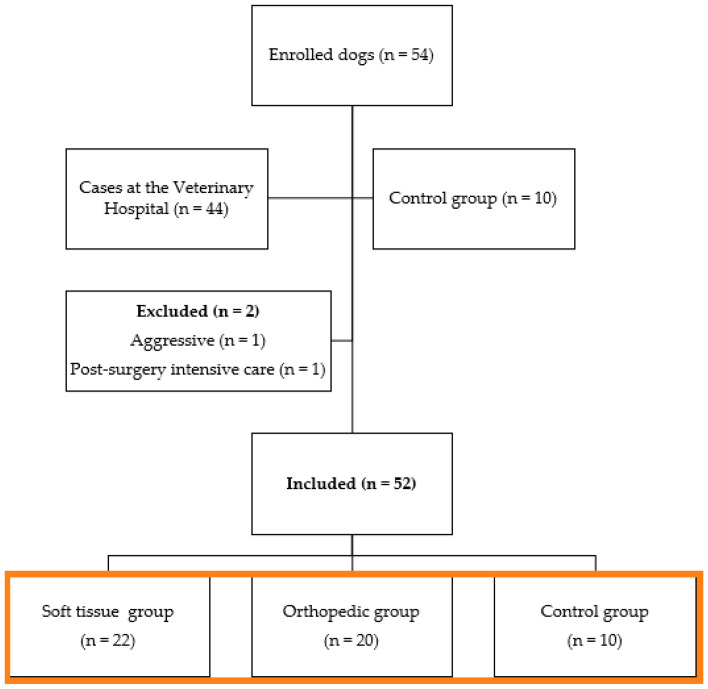
Flowchart of dogs included in the study (orange box).

**Figure 2 animals-14-00831-f002:**
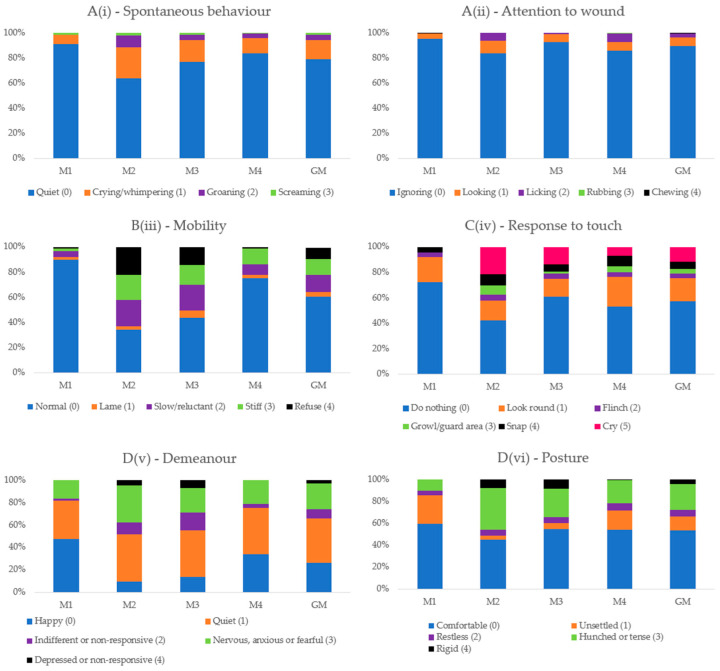
Frequency of the presence of scores of each item of the Short-Form Composite Pain Score. M1—basal, before surgery; M2—peak of pain, after surgery; M3—1 h after peak of pain and analgesia; M4—24 h after surgery; GM—Data of all time points together (M1 + M2 + M3 + M4). B(iii)—Mobility was considered only for soft tissue surgery group.

**Figure 3 animals-14-00831-f003:**
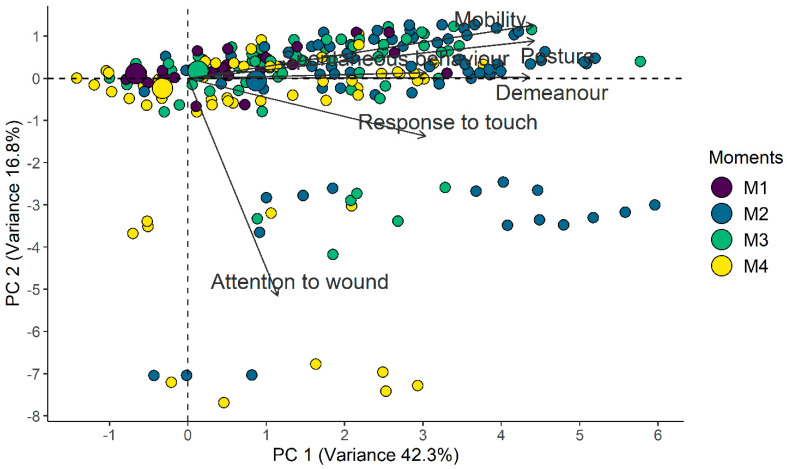
Biplot of the CMPS-SF principal component analysis. Smaller circles indicate each assessment while larger circles indicate the centroid of each timepoint and represent the gravitational center of all the vectors when a line is drawn from points of the same color, leading to its geometric center. Therefore, the centroid related to the time point of greatest pain (M2) is positioned to the right, as well as the positioning of the vectors of each item, while the centroids of least pain are positioned to the left (M1-baseline and M4-24 h), and the centroid for moderate pain (M3-rescue) is close to the center of the figure.

**Figure 4 animals-14-00831-f004:**
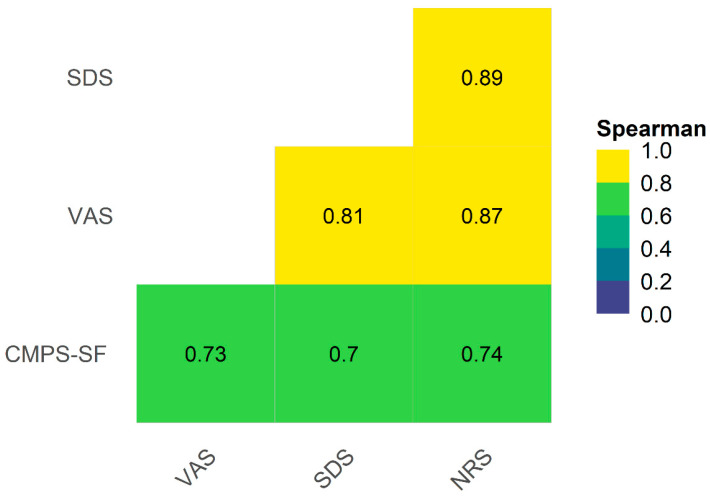
Heat map and correlogram between CMPS-SF, simple descriptive scale (SDS), numerical rating scale (NRS), and VAS. Interpretation: VAS—Visual analogue scale, CMPS-SF—Short-Form Composite Measure Pain Scale. Interpretation for Spearman correlation coefficient < 0.19—very weak, 0.2–0.39—weak, 0.4–0.59—moderate, 0.6–0.79—strong, and 0.8–1—very strong [30].

**Figure 5 animals-14-00831-f005:**
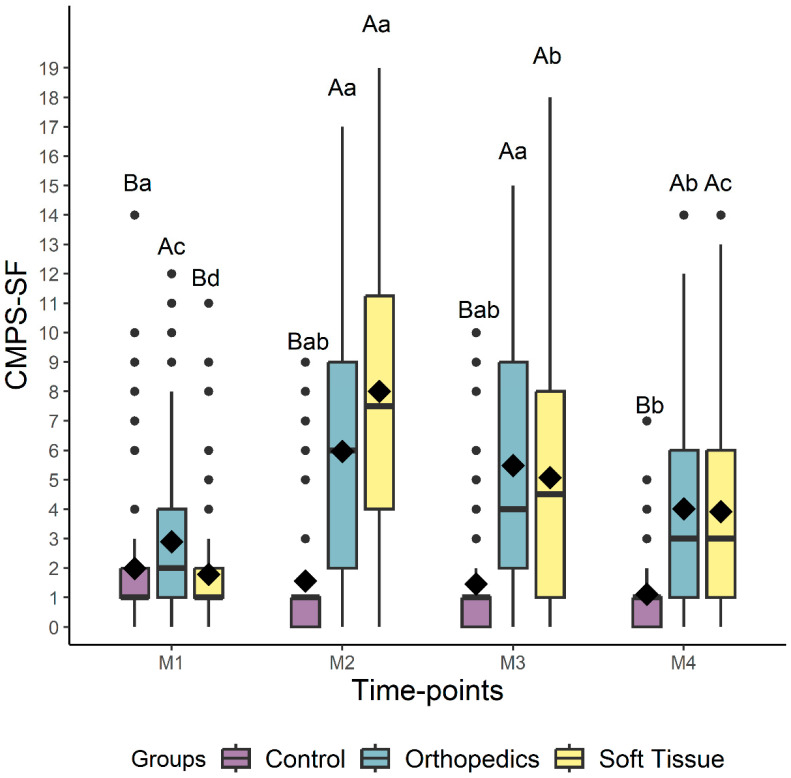
Box-plots of the scores of the CMPS-SF (Short-Form Composite Measure Pain Scale) comparing the perioperative time points for dogs submitted to control, soft tissue surgery, and orthopedic surgery separately. The top and bottom box lines represent the interquartile range (25 to 75%), the bold line within the box represents the median, the extremes of the whiskers represent the minimum and maximum values, black diamonds (♦) represent the mean, and black circles (●) represent outliers. Different lowercase letters indicate significant differences between time points (a > b > c), and capital letters indicate differences between groups at each time point. M1 is before surgery (basal), M2 is peak of pain (post-surgery), M3 is 1 h after peak of pain (rescue analgesia), and M4 is 24 h after surgery.

**Figure 6 animals-14-00831-f006:**
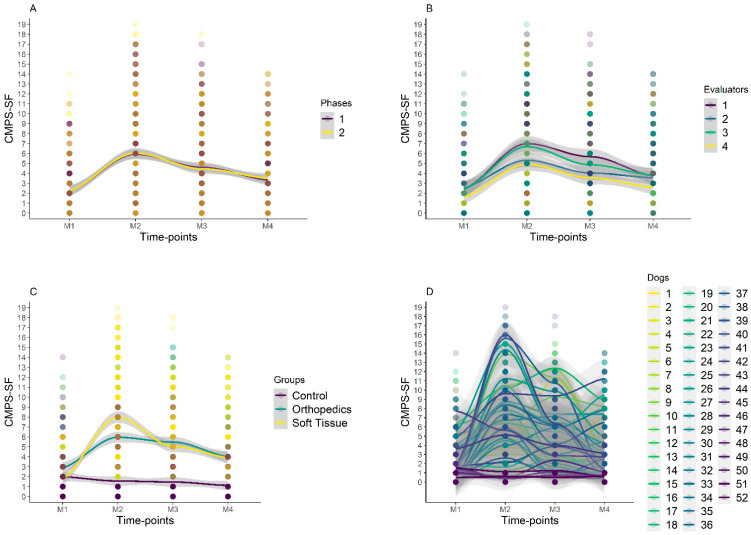
Smooth tendency lines, according to the LOESS method, indicating the scores of CMPS-SF over timepoints and for each phase (**A**), evaluators (**B**), groups (**C**), and dogs (**D**). The shaded area corresponds to the standard error of the smooth lines, M1 is before surgery (basal), M2 is peak of pain (post-surgery), M3 is 1 h after peak of pain (rescue analgesia), and M4 is 24 h after surgery. Dogs 1–20 is orthopedic group, 21–42 is soft tissue group, and 43–52 is control group. Graph A shows that there were no variations between phases 1 and 2 for all evaluators and time points. Graph C shows higher scores for orthopedic and soft tissue groups at M2, with a reduction in scores at M3 and a further reduction at M4. There was no variation in CG scores between time points.

**Figure 7 animals-14-00831-f007:**
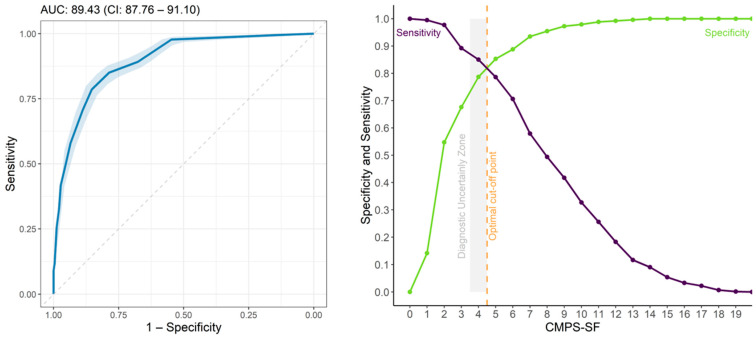
ROC curve and AUC (**left**), two-graph ROC curve with the zone of diagnostic uncertainty for CMPS-SF (**right**) for all groups with 95% confidence interval (CI), calculated from 1001 replications and area under the curve (AUC) (**left**). Interpretation: ROC Curve is Receiver Operating Characteristic, Discriminatory capacity-AUC: 0.5–0.7, low accuracy; 0.71–0.9, moderate; AUC > 0.9, high accuracy [34].

**Table 1 animals-14-00831-t001:** Intra-observer reliability of four raters for indication of analgesic rescue, visual analogue scale, simple descriptive scale, numerical rating scale, and CMPS-SF in the perioperative period of dogs undergoing orthopedic (n = 20) or soft tissue (n = 22) surgeries and negative controls (n = 10).

Statistical Test	Item or Sub-Item	PhD Student	Ethologist	MSc Student	Student
Kw (CI)	Analgesic rescue	**0.71 (0.61–0.8)**	**0.93 (0.88–0.98)**	**0.84 (0.75–0.92)**	**0.94 (0.89–0.99)**
	Numerical rating scale	**0.83 (0.78–0.87)**	**0.82 (0.72–0.88)**	**0.93 (0.87–0.97)**	**0.91 (0.85–0.94)**
	Simples descriptive scale	**0.76 (0.68–0.83)**	**1 (1–1)**	**0.93 (0.87–0.97)**	**0.88 (0.82–0.93)**
ICC (CI)	Visual analogue scale	**0.91 (0.88–0.93)**	**0.88 (0.85–0.91)**	**0.85 (0.8–0.9)**	**0.94 (0.92–0.95)**
Short-Form Composite Pain Scale (CMPS-SF)					
Kw (CI)	A(i)—Spontaneous behavior	0.56 (0.41–0.69)	**0.7 (0.52–0.83)**	**0.88 (0.79–0.95)**	**0.94 (0.88–0.99)**
	A(ii)—Attention to wound	**0.62 (0.33–0.81)**	**0.67 (0.32–0.88)**	**0.65 (0.46–0.78)**	**1 (1–1)**
	B(iii)—Mobility	**0.71 (0.57–0.81)**	**0.76 (0.63–0.87)**	**0.91 (0.82–0.98)**	**0.98 (0.95–1)**
	C(iv)—Response to touch	**0.74 (0.62–0.84)**	**0.8 (0.7–0.88)**	**0.82 (0.72–0.91)**	**1 (0.99–1)**
	D(v)—Demeanor	**0.74 (0.67–0.8)**	0.37 (0.24–0.49)	**0.7 (0.6–0.78)**	**0.92 (0.87–0.99)**
	D(vi)—Posture	**0.61 (0.5–0.72)**	0.51 (0.38–0.63)	**0.53 (0.4–0.65)**	**0.94 (0.87–0.99)**
ICC (CI)	Total	**0.91 (0.88–0.93)**	**0.85 (0.8–0.88)**	**0.93 (0.9–0.94)**	**0.99 (0.99–0.99)**

Interpretation: 0.81–1.0 very good; 0.61–0.80 good; 0.41–0.60 moderate; 0.21–0.40 reasonable; <0.20 poor [35]. CMPS-SF is Short-Form Composite Pain Score [10], Kw: Weighted Kappa coefficient, ICC: Intraclass correlation coefficient, CI: 95% confidence interval (minimum-maximum values). In bold are the values that fulfilled the adapted GRADE criteria [13,14].

**Table 2 animals-14-00831-t002:** Loading values, eigenvalues, variance, and cumulative variance of the CMPS-SF [10] items based on principal component analysis (n = 52 dogs).

Items	Principal Component 1	Principal Component 2
A(i)—Spontaneous behavior	**0.55**	0.02
A(ii)—Attention to wound	0.21	**−0.93**
B(iii)—Mobility	**0.80**	0.23
C(iv)—Response to touch	**0.55**	−0.25
D(v)—Demeanor	**0.79**	0.00
D(vi)—Posture	**0.80**	0.16
Eigenvalue	**2.54**	**1.01**

Interpretation: CMPS-SF—Short-Form Composite Measure Pain Score [10]. Items with a loading value ≥ 0.50 or ≤0.50 (highlighted in bold) were considered to have a representative dimension (eigenvalue > 1 and variance > 20%) [36].

**Table 3 animals-14-00831-t003:** Confirmatory factor analysis using all CMPS-SF items in a unidimensional structure in the perioperative period of 52 dogs undergoing orthopedic (n = 20) or soft tissue (n = 22) surgeries and negative controls (n = 10).

	CMPS-SF
Comparative Fit Index	0.96
Tucker–Lewis Index	0.93
Root Mean Square Error of Approximation	0.073 (0.056–0.092)

Interpretation for Comparative Fit and Tucker–Lewis Indexes: good > 0.90–0.94, excellent ≥ 0.95. Root Mean Square Error of Approximation acceptable values < 0.05 [37].

**Table 4 animals-14-00831-t004:** Median (amplitude) [interquartile range] of the analgesic rescue (RA), simple descriptive scale (SDS), numerical rating scale (NRS), visual analogue scale (VAS), CMPS-SF, and its items over time points. (n = 52 dogs).

		Median (Amplitude) [Interquartile Range]				Significance in the Model Effects			
Items	Groups	M1	M2	M3	M4	Evaluator	Seq:Timepoints	Phase	Sequence
RA	CG	0 (0–1) [0–0] Ba	0 (0–1) [0–0] Ba	0 (0–1) [0–0] Ba	0 (0–0) [0–0] Aa	*	**		
	OG	0 (0–1) [0–1] Ac	1 (0–1) [0–1] Aa	1 (0–1) [0–1] Ab	0 (0–1) [0–1] Ab				
	STG	0 (0–1) [0–0] Bc	1 (0–1) [0–1] Aa	0 (0–1) [0–1] Ab	0 (0–1) [0–1] Ab				
NRS	CG	1 (1–7) [1–1] Ba	1 (0–5) [1–1] Ba	1 (1–5) [1–1] Ba	1 (1–4) [1–1] Ba	*	*		
	OG	2 (1–8) [1–3] Ac	3 (1–9) [2–6] Aa	3 (1–10) [1–5] Aab	2 (1–9) [1–4] Ab				
	STG	1 (0–7) [1–1] Bd	3.5 (1–10) [2–6] Aa	2 (1–10) [1–5] Ab	1 (1–9) [1–3] Ac				
SDS	CG	1 (1–3) [1–1] Ba	1 (0–2) [1–1] Ba	1 (1–2) [1–1] Ba	1 (1–2) [1–1] Ba	*			
	OG	1 (1–4) [1–2] Ab	2 (1–4) [1–3] Aa	2 (1–4) [1–3] Aa	2 (1–4) [1–2] Aab				
	STG	1 (1–3) [1–1] Bc	2 (1–4) [1–3] Aa	1 (1–4) [1–2] Aab	1 (1–4) [1–2] Ab				
VAS	CG	0 (0–41) [0–4] Ba	0 (0–38) [0–4] Ba	0 (0–38) [0–4] Ba	0 (0–20) [0–4] Ba	*	*		
	OG	13 (0–63) [4–22] Ab	25 (0–80) [10–45] Aa	17 (0–83) [4–44] Aa	17 (0–76) [4–33] Aab				
	STG	0 (0–53) [0–4] Bc	28 (0–83) [8.5–49] Aa	11 (0–83) [1–33] Ab	6 (0–81) [0–25] Ab				
A(i)—Spontaneous behavior	CG	0 (0–1) [0–0] Ab	0 (0–0) [0–0] Ab	0 (0–0) [0–0] Ab	0 (0–3) [0–0] Aa	**			
	OG	0 (0–3) [0–0] Ab	0 (0–3) [0–1] Aa	0 (0–3) [0–1] Aa	0 (0–3) [0–1] Aa				
	STG	0 (0–3) [0–0] Aab	0 (0–3) [0–1] Aa	0 (0–3) [0–0] Ab	0 (0–2) [0–0] Bc				
A(ii)—Attention to wound	CG	0 (0–0) [0–0] Aa	0 (0–0) [0–0] Aa	0 (0–1) [0–0] Ba	0 (0–0) [0–0] Ba	*	*		
	OG	0 (0–4) [0–0] Ac	0 (0–2) [0–0] Aa	0 (0–2) [0–0] Abc	0 (0–3) [0–0] Aab				
	STG	0 (0–0) [0–0] Aab	0 (0–2) [0–0] Aa	0 (0–1) [0–0] Bb	0 (0–2) [0–0] Bab				
B(iii)—Mobility	CG	0 (0–4) [0–0] Aa	0 (0–3) [0–0] Ba	0 (0–3) [0–0] Ba	0 (0–3) [0–0] Ba	***	**		*
	OG	--	--	--	--				
	STG	0 (0–4) [0–0] Ac	2 (0–4) [0–3] Aa	1.5 (0–4) [0–3] Aa	0 (0–4) [0–0.5] Ab				
C(iv)—Response to touch	CG	0 (0–5) [0–0] Ba	0 (0–2) [0–0] Ba	0 (0–5) [0–0] Ba	0 (0–2) [0–0] Ba		**		***
	OG	0 (0–5) [0–1] Ab	1 (0–5) [0–4] Aa	1 (0–5) [0–4] Aa	1 (0–5) [0–1] Ab				
	STG	0 (0–5) [0–0] Bd	0 (0–5) [0–4] Aa	0 (0–5) [0–0] Bc	0 (0–5) [0–3] Ab				
D(v)—Demeanor	CG	0 (0–3) [0–1] Ba	0 (0–3) [0–1] Bab	0 (0–3) [0–1] Ba	0 (0–3) [0–0] Bb	***	*	*	**
	OG	1 (0–3) [0–1] Ab	1 (0–4) [1–3] Aa	2 (0–4) [1–3] Aa	1 (0–3) [1–2] Ab				
	STG	0 (0–3) [0–1] Bc	2 (0–4) [1–3] Aa	1 (0–4) [1–3] Aa	1 (0–3) [0–1] Ab				
D(vi)—Posture	CG	0 (0–3) [0–1] Aa	0 (0–3) [0–1] Ba	0 (0–3) [0–1] Ba	0 (0–3) [0–1] Ba	***	*		
	OG	0 (0–3) [0–1] Ab	0 (0–4) [0–3] Aa	0 (0–4) [0–3] Aa	1 (0–4) [0–3] Aa				
	STG	0 (0–3) [0–1] Ac	3 (0–4) [0–3] Aa	0 (0–4) [0–3] Ab	0 (0–4) [0–1] Abc				
Total score CMPS-SF	CG	1 (0–14) [1–2] Ba	1 (0–9) [0–1] Bab	1 (0–10) [0–1] Bab	1 (0–7) [0–1] Bb	**			
	OG	2 (0–12) [1–4] Ac	6 (0–17) [2–9] Aa	4 (0–15) [2–9] Aa	3 (0–14) [1–6] Ab				
	STG	1 (0–11) [1–2] Bd	7.5 (0–19) [4–11.5] Aa	4.5 (0–18) [1–8] Ab	3 (0–14) [1–6] Ac				

Interpretation: Different lowercase letters express significant differences between time points (a > b > c > d), and capital letters indicate differences between groups (A > B > C); *** is *p* < 0.001, ** is *p* < 0.01, and * is *p* < 0.05; ‘Seq: Timepoints’ is the interaction between sequence and timepoints as fixed effect in the models. CMPS-SF is Short-Form Composite Measure Pain Scale, CG is negative control group, OG is orthopedic group, STG is soft tissue group, M1 is before surgery, M2 is peak of pain post-surgery, M3 is 1h after the peak of pain and analgesia rescue, and M4 is 24 h after surgery.

**Table 5 animals-14-00831-t005:** Internal consistency, item-total correlation, and McDonald coefficient of the CMPS-SF (n = 52 dogs).

Item Tests	Item-Total (Spearman)	Cronbach’s Coefficient (α)	McDonald’s Coefficient (ω)
Complete CMPS-SF		0.70	0.77
	Excluding each item below		
A(i)—Spontaneous behavior	**0.34**	**0.69**	**0.76**
A(ii)—Attention to wound	0.21	**0.72**	**0.80**
B(iii)—Mobility	**0.58**	0.59	**0.71**
C(iv)—Response to touch	**0.33**	**0.71**	**0.76**
D(v)—Demeanor	**0.47**	**0.60**	**0.71**
D(vi)—Posture	**0.42**	0.59	**0.70**

Interpretation of Spearman correlation coefficient (Rho): 0.3–0.7 [31] (values in bold indicate scores ≥ 0.3 and ≤0.7). Interpretation of Cronbach’s α coefficient values: 0.60–0.64 minimally acceptable; 0.65–0.69 acceptable; 0.70–0.74 good; 0.75–0.80 very good; >0.80 excellent [31] (bold values indicate scores ≥ 0.6). For the coefficient w, acceptable values should be between 0.65 and 0.8 and strong evidence > 0.8. CMPS-SF is Short-Form Composite Measure Pain Scale.

**Table 6 animals-14-00831-t006:** Specificity and sensitivity of simple descriptive scale (SDS), numerical rating scale (NRS), visual analogue scale (VAS), and CMPS-SF items.

	Specificity (Control Group All Time Points)			Sensitivity (Surgical Groups at M2)		
	Estimated	Low CI	High CI	Estimated	Low CI	High CI
A(i)—Spontaneous behavior	**88**	**81**	**93**	59	54	63
A(ii)—Attention to wound	**98**	**90**	**100**	53	49	57
B(iii)—Mobility	**77**	**69**	**83**	**83**	**78**	**86**
C(iv)—Response to touch	**89**	**85**	**93**	68	63	**72**
D(v)—Demeanor	**78**	**73**	**82**	**88**	**83**	**91**
D(vi)—Posture	56	51	62	54	48	59
Sum of CMPS-SF	**83**	**78**	**87**	**79**	**74**	**83**
NRS	51	47	55	**100**	2	**100**
SDS	51	47	55	**100**	2	**100**
VAS	**90**	**85**	**93**	**75**	**70**	**79**

Interpretation: Excellent 95–100%; good 85–94.9%; moderate 70–84.9%; non-specific or non-sensitive < 0.70. Values highlighted in bold ≥ 70%, Sensitivity is “S” and Specificity is “Sp”.

**Table 7 animals-14-00831-t007:** Threshold, specificity, sensitivity, and area under the curve (AUC) for CMPS-SF [10], simple descriptive scale (SDS), numerical rating scale (NRS), visual analogue scale (VAS) including all groups.

Scale	Excluding	AUC (Min–Max)	YI	Cut-Off Point	Sensitivity	Specificity
NRS	-	0.895 (0.88–0.91)	0.54	≥3 (2.5)	61	93
SDS	-	0.906 (0.89–0.92)	0.75	≥2 (1.5)	94	80
VAS	-	0.903 (0.89–0.92)	0.73	≥17 (16.5)	91	82
CMPS-SF (STG + OG)	-	0.89 (0.88–0.91)	0.64	≥5 (4.5)	78	85
STG + OG	A(ii)—Attention to wound	0.89 (0.87–0.9)	0.64	≥4 (3.5)	84	80
	B(iii)—Mobility	0.89 (0.88–0.91)	0.65	≥4 (3.5)	83	82
	A(ii)—Attention to wound + B(iii)—Mobility	0.87 (0.85–0.9)	0.61	≥4 (3.5)	80	81
	C(iv)—Response to touch	0.8 (0.775–0.825)	0.504	≥4 (3.5)	68	82
OG	B(iii)—Mobility	0.87 (0.85–0.9)	0.6	≥5 (4.5)	72	88
	A(ii)—Attention to wound + B(iii)—Mobility	0.86 (0.84–0.89)	0.58	≥4 (3.5)	77	81
	C(iv)—Response to touch	0.768 (0.73–0.8)	0.475	≥3 (2.5)	72	76
STG	Full scale	0.92 (0.9–0.94)	0.7	≥5 (4.5)	87	84
	A(ii)—Attention to wound	0.91 (0.89–0.94)	0.71	≥4 (3.5)	92	79
	B(iii)—Mobility	0.91 (0.89–0.93)	0.71	≥4 (3.5)	88	83
	A(ii)—Attention to wound + B(iii)—Mobility	0.91 (0.88–0.93)	0.71	≥4 (3.5)	88	84
	C(iv)—Response to touch	0.836 (0.803–0.87)	0.59	≥5 (4.5)	73	86

Interpretation: AUC is area under the curve, OG is orthopedic group, STG is Soft Tissue Group, YI is Youden Index.

**Table 8 animals-14-00831-t008:** Percentage of assessments in the CMPS-SF diagnostic uncertainty zone (4).

Evaluator	M1	M2	M3	M4
PhD student	7%	5%	13%	7%
Ethologist	7%	6%	6%	7%
MSc student	3%	7%	9%	9%
Student	0%	6%	0%	8%
Mean	4%	6%	7%	7%

## Data Availability

Data are contained within the article and Appendix A.

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
