# Peer review of "Validation of the Portuguese Version of the Short-Form Glasgow Composite Measure Pain Scale (CMPS-SF) According to COSMIN and GRADE Guidelines"

_animals, 2024, doi:10.3390/ani14060831_

Round 1

Reviewer 1 Report

Comments and Suggestions for Authors

This article deals with a very interesting topic for current clinical practice in veterinary medicine. I must congratulate the authors as they have carried out a thorough review of the patients. However, it has several mistakes and errors that should be corrected (specified below). On the other hand, from my point of view the manuscript is too long and hard to follow sometimes. I think the use of acronyms, although appropriate, makes reading very difficult, especially when talking about observers.

- Lines 98-105: I will try to move this paragraph to the Results section. You could explain that composition of the trial, however it is not appropiate to detail all the descriptive data of population.

- Lines 114-115: In my opinion, it would be better to place it next to the exclusion criteria.

- Line 129: I suppose numver 23 should be write in superscript number.

- In section 2.6: I don´t really understand why if patients should have an intravenous catheter for the anaesthesia of the surgery, why did you administer the rescue drugs intramuscularly?

- In section 2.10, could you explain me the reasons why you choose an interval of 21 days between assessments?

- In section 2.11.1, I don´t unsderstand why do you use the chi-square test if you said before that the data had a no normal distribution.

- In section 2.11.3, could you reference which graphs does it correspond to and where are they?

- In Table 1: I would change, if possible, the initials of the authors/observers to the degree they have, to make reading easier.

Author Response

Dear Reviewer,

The authors appreciate the time and effort spent reviewing this manuscript and thank you very much for your comments. All corrections were performed according to your suggestions and each comment was responded separately.

            We hope that after these corrections you consider the manuscript suitable for publication, but we are happy to answer any further questions.

            Yours sincerely,

The authors

Reviewer #1:

This article deals with a very interesting topic for current clinical practice in veterinary medicine. I must congratulate the authors as they have carried out a thorough review of the patients. However, it has several mistakes and errors that should be corrected (specified below). On the other hand, from my point of view the manuscript is too long and hard to follow sometimes. I think the use of acronyms, although appropriate, makes reading very difficult, especially when talking about observers.

Answer: thank you for your positive comments. The corrections were performed as requested. We have shortened the manuscript and made it more fluent. Introduction was shortened and directed to the scope of the study. We do agree that the methods and results provide a lot of information which may not be simple to readers. We did our best to simplify but according to COSMIN and GRADE guidelines all these detailed descriptions are necessary to fit their requirements, otherwise the scientific level classification may be compromised. The discussion was streamlined to follow your suggestion. The acronyms were reviewed and most of them removed.

- Lines 98-105: I will try to move this paragraph to the Results section. You could explain that composition of the trial, however it is not appropriate to detail all the descriptive data of population.

Answer: this paragraph was moved to Results section as requested (pages 7-8 , line 313-323).

- Lines 114-115: In my opinion, it would be better to place it next to the exclusion criteria.

Answer: this part was moved to the exclusion criteria. (page 3, lines 111-112)

- Line 129: I suppose number 23 should be write in superscript number.

Answer: corrected

- In section 2.6: I don´t really understand why if patients should have an intravenous catheter for the anaesthesia of the surgery, why did you administer the rescue drugs intramuscularly?

Answer: this was explained in discussion (page 7, lines 635-636). Intramuscular injection was performed to minimize morphine-induced histamine release compared to the intravenous route.

- In section 2.10, could you explain me the reasons why you choose an interval of 21 days between assessments?

Answer: the rationale why this interval was chosen was included in section 2.10 “to prevent observers from remembering the first answer.” (page 5, lines 198-199)

- In section 2.11.1, I don´t understand why do you use the chi-square test if you said before that the data had a no normal distribution.

Answer: Data had no normal distribution for the total score of CMPS-SF, however chi-square does not have relation with data distribution. According to our statistician chi-square is the correct statistical test to compare the number of dogs that required rescue analgesia between the orthopedic versus soft tissue surgery groups.

- In section 2.11.3, could you reference which graphs does it correspond to and where are they?

Answer: This information is shown in Figure 2 as described in Results section (page 10, between lines 355-356)

- In Table 1: I would change, if possible, the initials of the authors/observers to the degree they have, to make reading easier.

Answer: Done

Reviewer 2 Report

Comments and Suggestions for Authors

Dear authors

The paper has some rationale, very interesting. I read the paper with pleasure. I have no major changes to ask for. The English writing is fluent. The focus is quite challenging in my opinion and it is not easy to be comprehensive and make it usable for Animals' target readers. In fact, the materials and methods and results sections are very information-laden. The discussions could probably be streamlined.

In any case, good work

Author Response

Dear Reviewer,

The authors appreciate the time and effort spent reviewing this manuscript and thank you very much for your comments. All corrections were performed according to your suggestions and each comment was responded separately.

            We hope that after these corrections you consider the manuscript suitable for publication, but we are happy to answer any further questions.

            Yours sincerely,

The authors

Reviewer #2:

Dear authors

The paper has some rationale, very interesting. I read the paper with pleasure. I have no major changes to ask for. The English writing is fluent. The focus is quite challenging in my opinion and it is not easy to be comprehensive and make it usable for Animals' target readers. In fact, the materials and methods and results sections are very information-laden. The discussions could probably be streamlined.

In any case, good work

Answer: Thanks for your positive comments. The introduction was shortened and directed to the scope of the study. We do agree that the methods and results provide a lot of information that may not be simple to readers. We did our best to simplify but according to COSMIN and GRADE guidelines all these detailed descriptions are necessary to fit their requirements, otherwise the scientific level classification may be compromised. The discussion was streamlined to follow your suggestion and the conclusion was better matched to results.

Reviewer 3 Report

Comments and Suggestions for Authors

The main question addressed by the research is whether the  assessment of pain is specific and sensitive,
The topic is relevant in the field, and 2. Do you consider the topic original or relevant in because we must reduce pain in animals as much as possible.
3. It adds to the subject area compared with other published
material by validating the method with several  evaluators ?
4. What specific improvements could the authors consider regarding the
methodology? Would like to know exactly which signs the observers were using to evaluate pain. The conclusions are consistent with the evidence and arguments
presented, and they address the main question posed.
The references are appropriate?

Minor comments 

77 "define the level of evidence"  not sure what that means 

 101 healthy9 remove 9

129 9,19–2123. what is 23?

venoclysis? venapuncture

Method

 134 How did you prevent the preoperative animals from eating?

144 invasive?

Does dashes for mobility mean that those dogs did not move?

455 modified the not modified de 

459  important  to assess pain  should be able 8 e important in assessing pain 

Although she 623

"might be able recognize the group pertaining each dog," Don't understand what this means. She could recognize which group ( soft tissue, orthopedics or control) ???

Comments on the Quality of English Language

See above 

Author Response

Dear Reviewer,

The authors appreciate the time and effort spent reviewing this manuscript and thank you very much for your comments. All corrections were performed according to your suggestions and each comment was responded separately.

Introduction was shortened and directed to the scope of the study. We included few details in method. This section follows COSMIN and GRADE guidelines. The discussion was streamlined, and conclusion was better matched to results.

            We hope that after these corrections you consider the manuscript suitable for publication, but we are happy to answer any further questions.

            Yours sincerely,

The authors

Reviewer #3:

Comments and Suggestions for Authors

The main question addressed by the research is whether the assessment of pain is specific and sensitive. The topic is relevant in the field,

Answer: according to the conclusion of the study the CMPS-SF is reliable and effective for diagnosing acute pain in dogs. The item-total correlation, internal consistency, responsiveness, specificity, and sensitivity ensured its content, criterion, and construct validity in accordance with COSMIN guidelines. The GRADE method provided evidence that our methodology increased the level of evidence of the scale. However, it is necessary to carry out a new statistical analysis after refinement to investigate whether the level of evidence could be increased to high instead of good.

  1. Do you consider the topic original or relevant in because we must reduce pain in animals as much as possible.

Answer: Yest that is our goal. That is why we developed the website www.animalpain.org and the application Vetpain® for the iOs https://apps.apple.com/br/app/vetpain/id6462712970

and Android https://play.google.com/store/apps/details?id=com.vetpain.app&pli=1

  1. It adds to the subject area compared with other published material by validating the method with several evaluators?

Answer: according to recommendation from the literature reference 24), to measure reliability, at least three evaluators should assess the proposed instrument. This approach has been previously performed for other pain scales in animals (please see the references 27, 28, 42 and 45)

  1. What specific improvements could the authors consider regarding the methodology?

Answer: We used the methods adapted from COSMIN guidelines which we believe is the best approach to investigate the level of scientific evidence of CMPS-SF. The possible limitations were included in discussion section. We believe that CMPS-SF requires refinement to improve its level of evidence and this was included in discussion.

Would like to know exactly which signs the observers were using to evaluate pain.

Answer: According to the methods section the evaluators assessed the CMPS-SF reported in reference 10

The conclusions are consistent with the evidence and arguments presented, and they address the main question posed.
Answer: Thanks for your positive comment.

Minor comments 

77 "define the level of evidence"  not sure what that means 

Answer: The level of evidence represents the quality of the available scientific evidence in relation to reliability and validation of the instrument. This information is available in the references 11, 12, 13 and 14 and in the Supplementary information 1

All other minor corrections were performed.

venoclysis? Venipuncture

Answer: corrected

Method

134 How did you prevent the preoperative animals from eating?

Answer: Food was withdrawn after the baseline videos were recorded and water was not avail-able for 2 hours before anesthesia (page 3; lines 125-126)

144 invasive?

Answer: amended to “invasively by intra-arterial catheterization”

Does dashes for mobility mean that those dogs did not move?

Answer: in these cases, mobility could not be assessed because dogs submitted to orthopedic surgery could not be moved.

455 modified the not modified de 

Answer: corrected

459 important  to assess pain  should be able 8 e important in assessing pain

Answer: corrected

Although she "might be able recognize the group pertaining each dog," Don't understand what this means. She might be able recognize the group pertaining each dog (soft tissue, orthopedics or control) ???

Answer: The sentence has been rephrased (page 22, lines 624-626)